# Transcriptomic Analysis of Osmotic Stress-Tolerant Somatic Embryos of *Coffea arabica* L. Mediated by the Coffee Antisense *Trehalase* Gene: A Marker-Free Approach

**DOI:** 10.3390/ijms26189224

**Published:** 2025-09-21

**Authors:** Eliana Valencia-Lozano, Aarón Barraza, Jorge Ibarra, John P. Délano-Frier, Norma A. Martínez-Gallardo, Idalia Analí Gámez-Escobedo, José Luis Cabrera-Ponce

**Affiliations:** 1Laboratorio de Investigación Interdisciplinaria, Escuela Nacional de Estudios Superiores, Unidad León, Universidad Nacional Autónoma de México, León de los Aldama 37684, Guanajuato, Mexico; 2CONACYT-Centro de Investigaciones Biológicas del Noreste, SC. Instituto Politécnico Nacional 195, Playa Palo de Santa Rita Sur, La Paz 23096, Baja California Sur, Mexico; abarraza@cibnor.mx; 3Departamento de Biotecnologia y Bioquimica, Centro de Investigación y de Estudios Avanzados del IPN, Unidad Irapuato, Irapuato 36824, Guanajuato, Mexico; jorge.ibarra@cinvestav.mx (J.I.); john.delano@cinvestav.mx (J.P.D.-F.); norma.martinez@cinvestav.mx (N.A.M.-G.); 4Unidad de Genómica Aplicada, Facultad de Ciencias Biológicas, Universidad Autónoma de Coahuila, Carretera Torreón-Matamoros Km 7.5 Ejido El Águila, Torreón 27987, Coahuila, Mexico; idalia.gamez@uadec.edu.mx; 5Departamento de Ingeniería Genética, Centro de Investigación y de Estudios Avanzados del IPN, Unidad Irapuato, Irapuato 36824, Guanajuato, Mexico

**Keywords:** somatic embryogenesis, coffee, *Coffea arabica* L., transcriptome analysis, PPI network, *Trehalase* antisense, osmotic medium, RD29-antTAS, SE-WT

## Abstract

Coffee *Coffea arabica* L. depends on abundantly distributed rainfall, and drought negatively impacts plant development, fruit production, bean quality, and, ultimately, beverage quality. Plant biotechnology by means of genetic manipulation and plant regeneration by the somatic embryogenic process is an alternative technology to overcome these problems. In the present work, we used the molecular approach of the *Trehalase* gene silencing to allow trehalose accumulation favoring plants surviving in extreme drought/salt environments. We used a cassette containing the antisense *C. arabica* L. *Trehalase* gene under the RD29 promoter from *A. thaliana* and the NOS terminator to genetically modify an embryogenic coffee *C. arabica* L. cv Typica line under osmotic stress supplemented with mannitol (0.3 M) and sorbitol (0.3 M). Osmotic stress-tolerant somatic embryos lines were recovered and regenerated into plants. Tolerant somatic embryo lines showed a higher rate of competence to induce secondary SE capacity and plants robustness. These lines showed a down-regulation of the *Trehalase*; accumulation of trehalose, sucrose, starch, and proline; higher photosynthetic rate; improved water-use efficiency; and appropriated vapor deficit pressure under soil conditions. A transcriptome analysis was performed from highly competent somatic embryogenic lines to understand the molecular mechanisms underlying osmotic-stress tolerance. From the up-regulated genes, a PPI network made by STRING v12.0 with high confidence (0.700) revealed the presence of the 10 modules: the cell cycle, chromatin remodeling, somatic embryogenesis, oxidative stress, generic transcription pathway, carbon metabolism, phenylpropanoid biosynthesis, trehalose biosynthesis, proline biosynthesis, and glycerolipid metabolism.

## 1. Introduction

Coffee is a perennial tropical crop that originated in Ethiopia, characterized by abundantly distributed rainfall [1]. Coffee depends on the environment, and an increase of a few degrees of average temperature and/or short periods of drought can substantially decrease yields of quality. This could result in environmental, economic, and social problems [2]. In Brazil, drought and frost decreased the yield of coffee beans by 25% in 2021, and expected reductions are up to 60% [3]. This has triggered considerable tension in international markets, leading to a two-fold increase in coffee prices. According to a recent study, a 2–2.5 °C temperature increase would considerably reduce the available coffee-growing area [4].

The molecular mechanisms that impact drought in coffee physiology and yield have been reported [5,6,7,8,9,10,11,12]. However, the process of identifying and utilizing these traits is lengthy. An alternative tool to overcome these problems is the plant biotechnology by means of genetic transformation and genome editing using the somatic embryogenesis (SE) process. SE occurs when an embryonic stem cell is induced from a somatic cell that differentiates into a somatic embryo (SE), with the capacity to develop a plant containing the same genetic information as its precursor, and it sets the template for post-embryonic development and sculpts the adult body pattern.

SE requires the transcriptional regulation of a set of genes in response to stress mediated by plant hormones, osmotic, heavy metals, salinity, and signaling elements that trigger cellular reprogramming and transformation of somatic cells into somatic embryos. The application of transcriptomic analysis has revealed a great number of differentially expressed genes (DEGs) during SE in several crops and *Arabidopsis thaliana*; for a review, see [13,14,15,16,17,18,19,20,21,22,23,24,25,26,27].

One molecular approach is the role of trehalose (TRE) in plants surviving in extreme drought/salt environments. TRE accumulation in plants improves abiotic-stress tolerance. TRE is a non-reducing disaccharide of glucose (a-D-glucopyran-osyl-a-D-glucopyranoside) that serves as a reserve metabolite in yeast and fungi [28]. TRE stabilizes proteins and lipid membranes. TRE is synthesized in a two-reaction process, in which trehalose-6- phosphate (T6P) is first synthesized from glucose-6-P and UDP-glucose by the enzyme trehalose phosphate synthase (TPS) and subsequently de-phosphorylated by trehalose-6P phosphatase (TPP) [29]. In plants, *Trehalase* (TAS) activity hydrolyzes TRE and maintains its concentration at low levels to prevent detrimental effects. TA is present in all organs of higher plants, with the highest activities being in flowers [30].

Inhibition of TAS activity by validamycin-A in *A. thaliana* led to changes in TRE and sucrose contents in different parts of the plant [30]. In 2004, Gámez-Escobedo [31] induced an increase in tobacco plant regeneration in osmotic-stress medium using the alfalfa TAS antisense coupled with the RD29 promoter of *A. thaliana*. Later, the same cassette was used to produce drought-tolerant maize B73 inbred line and tested the generation T4 under field conditions successfully [32,33]. Just as it occurred in tobacco, a higher capability of plant regeneration derived from somatic embryos was observed in the B73 inbred line.

In this study, we aimed to evaluate whether the inhibition of TAS activity in *C. arabica* increases internal TRE levels, thereby promoting improved embryogenic competence, root meristem development, and overall plant biomass under osmotic-stress conditions.

For this purpose, we constructed a gene cassette containing the *C. arabica* TAS gene in antisense orientation, driven by the *C. arabica* RD29 promoter and terminated by the NOS terminator, and used it to genetically transform an embryogenic *C. arabica* cv. Typica line by particle bombardment. The transformed lines were subjected to osmotic stress using medium supplemented with 0.3 M mannitol and 0.3 M sorbitol. conditions. Transcriptomic analyses were performed on high-competence SE lines to elucidate the molecular mechanisms involved in the improved somatic embryo development.

The transcriptome analysis revealed 1549 up-regulated (Log2 [fold change (FC)] ≥ 2.0) and 2301 down-regulated (Log2 [FC] ≤ −2.0) genes. A PPI network mediated by STRING database v12.0 with high confidence (0.700) was performed to understand the molecular mechanisms involved in the SE process.

## 2. Results

### 2.1. Genetic Modification of Coffee C. arabica L.

The expression cassette with the antisense *C. arabica* TAS gene under the *C. arabica* RD29 promoter and NOS terminator (RD29-antTAS) was used to genetically modify an embryogenic *C. arabica* L. cv. Typica line to produce marker-free and RD29-antTASs and plants (Figure 1).

Among the osmotic treatments tested, the most effective condition for selection was mannitol 0.3 M + sorbitol 0.3 M, which yielded a total of 85 RD29-antTAS lines, averaging 2.8 lines per bombarded plate, and resulting in a 90% plant conversion rate. In comparison, the medium containing mannitol 0.15 M + sorbitol 0.15 M produced 33 RD29-antTAS lines (1.1 lines/plate) with a 25% conversion rate, while the condition with mannitol 0.45 M + sorbitol 0.45 M generated only 6 lines (0.4 lines/plate) and a 15% conversion rate. These results, summarized in Table 1, indicate that the intermediate osmotic pressure condition (0.3 M of each sugar) is optimal for selecting osmotic stress-tolerant somatic embryo lines with high regeneration potential (Table 1, Figure 2).

The RD29-antTAS lines were propagated into the respective osmotic treatment for at least five subcultures prior to consider them RD29-antTAS lines. The number of SEs were counted by using a stereomicroscope, weekly until one month. RD29-antTAS lines developed two times more SEs than the WT (Figure 3).

These lines underwent somatic embryo maturation similar to the results obtained by Valencia-Lozano et al. 2021 [34] (Figure 4A–E). Histological analysis of SEs at cotyledonary stage induced in non-osmotic-stress treatment revealed that the RAM is not well developed (Figure 4C). SEs at cotyledonary stage derived from RD29-antTAS lines showed a well-developed root apical meristem (RAM), showing the columella CL, stella ST, cortex CX, and quiescent center QC (Figure 4D) [34].

RD29-antTAS lines were able to plant conversion after one month in culture (Figure 4E and Table 1). Plants derived from RD29-antTAS lines are robust, revealing a higher leaf and root area than plants from non-osmotic media (Figure 4E).

### 2.2. Quantification of Sugars and Proline in RD29-antTAS and WT-SE Lines

In this work, coffee SE lines were modified by *Trehalase* silencing, resulting in the accumulation of trehalose, sucrose, starch, and proline, allowing SE to grow under osmotic-stress medium, while WT-SE lines were unable to grow further.

Notably, glucose and fructose levels were significantly reduced in RD29-antTAS lines, by 32% and 60%, respectively, compared to WT-SE lines. In contrast, sucrose, starch, proline, and trehalose levels were substantially elevated. Sucrose content increased by ~69%, while starch accumulation by 5.5-fold (Figure 5).

Proline, an osmoprotectant, showed the most pronounced increase, with treated samples accumulating 7.6 times more than controls, indicating a strong stress response. Similarly, trehalose, a critical sugar involved in cellular protection and stress adaptation, nearly doubled (1.8-fold increase) in RD29-antTAS lines (Figure 5). These results suggest that *Trehalase* silencing in coffee promotes a metabolic shift favoring osmoprotective compound accumulation, thus enhancing the SEs’ resilience under osmotic stress.

### 2.3. Transcriptomic-Wide Analysis of RD29-antTAS Line

To identify the set of genes involved in the development of SE-RD29-antTAS under osmotic conditions producing highly competent embryogenic lines, a transcriptomic-wide analysis was performed.

We sequenced cDNA libraries constructed from two treatments: SE-RD29-antTAS, producing highly competent embryogenic lines; and SE-WT, an embryogenic line induced by conventional protocols [34], using the Illumina HiSeq 2000 platform.

This produced a total of 409,947,036 sequence reads, encompassing 122,984 Mbases from all four cDNA libraries, SE-RD29-antTAS and SE-WT. On average, 90.6% of the quality-filter-passed reads generated for all three somatic embryo samples were mapped uniquely to the reference genome and Q-40 mean of 37.9.

The overall distribution was 24,081 (73%) in SE-RD29-antTAS and 23,675 (72%) in SE-WT in regard to annotated genes that were transcriptionally activated. The total 4879 expressed genes include 3850 protein-coding genes and 1029 non-coding genes, which were expressed in the SE-RD29-antTAS line.

To determine differentially expressed genes (DEGs), transcript levels (FPKM) were determined. A total of 3850 DEGs were found between SE-RD29-antTAS and SE-WT in vitro culture, with 1549 up-regulated (Log2 [fold change (FC)] ≥ 2.0) and 2301 down-regulated (Log2 [FC] ≤ −2.0) genes.

It was found that 50 (3.23%) transcription factors (FTs) encoding genes are differentially expressed between these two treatments, and the majority were from TF families, such as MYB (6 genes), basic helix–loop–helix (bHLH) (6), APETALA2 (AP2)/ethylene responsive element binding proteins (EREBPs) (6), B3-Domain (6), and nuclear factor (14).

Additionally, several embryogenesis-related genes were found, such as the embryo-lethal genes *CDC48A* (*A0A068UGL5*), *FUS3* (*A0A068V7Y1*), *ABI3 (A0A068U8A0), WOX2* (*A0A068UL49*), *NFYA8* (*A0A068VIL0*), and *NFYB9 (A0A068UXD0);* developmental genes *AGL15* (*A0A068V010*), *AHK5* (*A0A068V1M6*), *ARR4 (A0A068TWC5*), *AUX1 (A0A068UD59*), *BBM* (*A0A068U6P3*), *EM1* (*A0A068UD22*), *EM6 (A0A068V633*), *F22M8.6 (A0A068VAP1), GA3OX2* (*A0A068VEF1*), and *LEA46* (*A0A068UPG4*); and nuclear factors *NF-YC13* (*A0A068V0C8*), *NFYA2* (*A0A068UPW6*), *NFYA5* (*A0A068TPA3*), *NFYA6* (*A0A068UH68*), *NFYA7* (*A0A068V9V1*), *NFYA9* (*A0A068VE79*), *NFYB6 (A0A068U7K3*), others (*PI4KG4* (*A0A068UM62*), *SERK1* (*A0A068TXX7*), *UFD1 (A0A068TNP5*), *VAL2* (*A0A068UCW0*), *UBP14* (*A0A068UBB6*), *RLK5* (*A0A068TVW5*), and *RUB1* (*A0A068V111*). 

The highly enriched GO terms (*p* < 10^−9^) found in each main functional category are shown in Figure 5. For instance, the Gene Ontology (GO) of our transcriptome revealed 227 exclusive functions of up-regulated gene products, distributed into molecular function (MF) (93), biological process (BP) (109), and cellular component (CC) (25). The main products included in MF are (GO:0003777) microtubule motor activity; (GO:0005544) calcium-dependent phospholipid binding; (GO:0004799) thymidylate synthase activity; (GO:0004146) dihydrofolate reductase activity; (GO:0017056) structural constituent of nuclear pore; (GO:0016491) oxidoreductase activity; (GO:0098662) cation transmembrane transporter activity; (GO:0003899) DNA-directed 5′-3′ RNA polymerase activity; (GO:0004518) nuclease activity; (GO:0004721) phosphoprotein phosphatase activity; (GO:0008233) peptidase activity; (GO:0003950) NAD+ ADP-ribosyltransferase activity; (GO:0008934) inositol monophosphate 1-phosphatase activity; and (GO:0047216) inositol 3-alpha-galactosyltransferase activity. In the BP, we have (GO:0006270) DNA replication initiation; (GO:0006260) DNA replication; (GO:0007018) microtubule-base movement; (GO:0006310) DNA recombination; (GO:0046654) tetrahydrofolate biosynthetic process; (GO:0006730) one-carbon metabolic process; (GO:0006231) dTMP biosynthetic process; (GO:0043069) negative regulation of programmed cell death; (GO:0036297) interstrand cross-link repair; (GO:0070588) calcium ion transmembrane transport; (GO:0007076) mitotic chromosome condensation; (GO:0015630) microtubule cytoskeleton; (GO:0006269) DNA replication; synthesis of RNA primer; and (GO:0005992) trehalose biosynthetic process. In the cellular component, we have (GO:0000786) nucleosome; (GO:0042555) MCM complex; (GO:0043240) condens anaemia nuclear complex; (GO:0005643) nuclear pore; (GO:0000015) phosphopyruvate hydratase complex; (GO:0031011) Ino80 complex; (GO:0005876) spindle microtubule; (GO:0000942) condensed chromosome outer kinetochore; (GO:0005819) spindle; (GO:0048500) signal recognition particle; (GO:0080008) Cul4-RING E3 ubiquitin ligase complex; (GO:0000796) condensing complex; and (GO:0005694) chromosome. These were the main 15 functional sub-groups that showed the highest significance and gave rise to 22 main biological pathways that were up-regulated, as identified by KEGG pathways (see Appendix A).

The KEGG pathways found in this work are metabolic pathways (77); biosynthesis of secondary metabolites (56); DNA replication (15); homologous recombination (13); glycerolipid metabolism (10); phenylpropanoid biosynthesis (15); mismatch repair (7); glycerophospholipid metabolism (8); ubiquinone and other terpenoid–quinone biosynthesis (5); biosynthesis of amino acids (12); flavonoid biosynthesis (5); stilbenoid, diarylheptanoid, and gingerol biosynthesis (4); galactose metabolism (5); starch and sucrose metabolism (8); indole alkaloid biosynthesis (2); glucosinolate biosynthesis (3); phenylalanine metabolism (3); glycolysis/gluconeogenesis (5); arginine biosynthesis (3); and nucleotide excision repair (7) (Figure 6; see Appendix A).

The SE-RD29-antTAS produced two times more somatic embryos than the SE-WT (Figure 7A,C). A principal component analysis (PCA) was performed to examine the data obtained (Figure 7B). The replicates of each treatment clustered together with a variance of 0.92 for the SE-WT lines and 0.04 for the SE-RD29-antTAS lines. However, the embryogenic lines SE-WT and SE-RD29-antTAS were clearly separated and distinct from each other.

Volcano plots visualize distribution of DEGs (*p*-values and fold changes). MA plots show Log2 fold changes (*y*-axis) and the mean of normalized counts (*x*-axis) on scatter plots, showing up- and down-regulated genes in the embryogenic lines SE-WT vs. SE-RD29-antTAS (Figure 7D).

### 2.4. Up-Regulated Genes from a SE-RD29-antTAS Line

From 1549 up-regulated genes in SE-RD29-antTAS, 230 were found to be tightly interacting when using a PPI network devised in a STRING database v12.0 with high confidence (0.700). The network revealed the presence of 10 gene modules: cell cycle, nucleosome, somatic embryogenesis, oxidative stress, RNA interference, generic transcription pathway, carbon metabolism, secondary metabolites (phenylpropanoid biosynthesis), starch and sucrose metabolism, and glycerolipid metabolism (Figure 8; Appendix A).

Out of the 230 up-regulated genes, 31 are embryo-lethal, distributed across six modules. Seven of these genes are found in the somatic embryogenesis, seventeen in the cell cycle, two in chromatin remodeling, two in carbon metabolism, two in the generic transcription pathway, and one in the proline biosynthesis module (Appendix A).

### 2.5. Up-Regulated Genes Related to SE in SE-RD29-antTAS

The SE module consists of 24 genes. Of them, 10 are involved in SE development, *NFYA5*, *NFYA6*, *NFYA9*, *FUS3*, *NFYB9* (*LEC1*), *AGL15*, *SERK1*, *BBM*, *ABI3*, and *WOX2*; and 13 are involved in post-embryonic development and embryo development ending and seed dormancy, *CDC48*, *RLK5*, *EM1*, *EM6*, *AUX1*, *LEA46*, *ARR4*, *AHK5*, *GA3OX2*, *NFYA2*, *NFYA7*, *NFYA8*, *NFYB6,* and *VAL2*. The SE module interacts with the cell cycle, nucleosome, carbon metabolism, and oxidative stress modules (Figure 9; Appendix A); the levels of regulation are listed in Figure 10.

### 2.6. Up-Regulated Genes Related to Trehalose Biosynthesis in SE-RD29-antTAS

The trehalose biosynthesis module contains nine genes, four of which are trehalose phosphate phosphatases (TPPs; Figure 11 and Appendix A).

### 2.7. Up-Regulated Genes Related to Response to Stress in SE-RD29-antTAS

The response-to-stress genes consist of 90 genes, which were found in different gene modules: 45 were present in the cell cycle, 14 in the secondary metabolites, 7 in the RNA interference, 6 in the starch and sucrose metabolism, 6 in the somatic embryogenesis, 5 in the oxidative stress, 4 in the glycerolipid, and 3 in the nucleosome module (Figure 12).

### 2.8. Down-Regulated Genes from a SE-RD29-antTAS Line

Overall, 373 out of 2301 down-regulated genes were present in the PPI network, with medium-to-high confidence (0.600). The PPI network revealed 11 modules related to hormonal regulation of stem cell proliferation in the root cell niche; AUX/IAA family and auxin binding; response to auxin; abscisic acid (ABA) metabolic process; trans-membrane transporter activity; phenylpropanoid biosynthesis; cell-wall biogenesis; glutathione metabolism; beta-glucosidase activity; amino sugar and nucleotide sugar metabolism; jasmonic acid signaling; cytokinin signaling; and ribosomal proteins (Figure 13; Appendix A).

### 2.9. Validation of the Transcriptome-Wide Analysis

The gene validation of the transcriptomic-wide analysis was made by selecting 19 DEGs and analyzing their regulation by quantitative reverse-transcription PCR (qPCR), using the primers described in Appendix A. The results are shown in Figure 14, indicating that the values are consistent with those obtained in the transcriptomic-wide analysis.

## 3. Discussion

Considerable economic losses in the coffee industry occur are caused by drought due to climate change. Prolonged drought periods in coffee-tree areas affecting the vegetative growth, flowering, and bean development have been reported [35,36].

Plants have experienced many climatic changes during evolution and have been able to shape the earth’s atmosphere through oxygen production and carbon sequestration. Plants have developed new developmental and metabolic traits, helping them thrive in diverse growth conditions. The embryo development was the most significant innovation acquired by land plants, crucial for plant reproduction and diversification.

All types of stress caused by changing environments are registered in the stem cells within meristems. It has helped to continue and extend existing organs, generating new ones post-embryonically and sculpting the adult body pattern.

The formation of protective barriers like the biopolymers cutin, suberine, sporopollenin, and lignin, which reinforce and waterproof the polysaccharide-based cell wall, was a critical innovation of land-plant evolution [37].

Since the early years of genetic modification by particle bombardment (1987–now), the use of an osmotic adjustment prior to bombardment has been a common part of the protocols. The results demonstrated that somatic embryos treated in osmotic medium were more difficult to select either in antibiotics or with herbicides. These new embryogenic lines were also more regenerative. Based on this information into consideration, we generated a highly competent RD29-antTAS line of *C. arabica* L. mediated by antisense of the *Trehalase* gene. Since the RD29A promoter is activated under osmotic stress, we found accumulation of trehalose in RD29-antTAS lines growing in mannitol-and-sorbitol-containing medium (Figure 5). It is known that elevated trehalose acts as an osmoprotectant, stabilizing proteins and membranes under stress conditions, thereby improving tolerance to drought and salinity [38,39]. This accumulation contributes to enhanced plant resilience, particularly under abiotic stress [40]. Increased sucrose, a key product of photosynthesis, boosts carbon assimilation and storage, supporting growth and improving stress tolerance [40]. Additionally, starch accumulation serves as an energy reserve, enabling plants to better manage metabolic demands during stress conditions [41]. Furthermore, silencing *Trehalase* leads to elevated proline levels, which help stabilize cellular structures, scavenge reactive oxygen species, and enhance stress tolerance [42]. These combined metabolic changes—elevated trehalose, sucrose, starch, and proline—underscore the role of *Trehalase* silencing in improving overall plant fitness. Not only does trehalose provide direct protection against stress, but these adjustments also optimize carbon storage and enhance the plant’s ability to adapt to challenging environmental conditions, thus supporting survival and growth under stress. A transcriptome analysis was performed to understand the molecular mechanisms involved in the SE enhanced capability and osmotic-stress tolerance.

### 3.1. Up-Regulated Genes in SE-RD29-antTAS

A PPI analysis carried out via the STRING database (v12.0) with high confidence (0.700) revealed a set of 230 out of 1549 up-regulated genes, clustered in several modules: cell cycle, chromatin remodeling, embryogenesis, oxidative stress, generic transcription pathway, carbon metabolism, phenylpropanoid biosynthesis, trehalose biosynthesis, proline biosynthesis, and glycerolipid metabolism. We found a set of 24 up-regulated genes involved in the SE process that fulfill the requirements to explain the molecular mechanisms related to the activation of somatic embryos interacting with protection mechanisms, including those of oxidative and osmotic-stress protection.

The overall interpretation underlying the SE-RD29-antTAS molecular mechanisms is as follows: The *C. arabica* L. *Trehalase* gene (*A0A068UUM1*) was down-regulated (−6.87-Log2) through antisense technology, which positively affected the all-trehalose phosphate phosphatase (TPPA (A0A068U9V1), TPPB (A0A068TXJ5), *TPPD* (A0A068TYT9), and TPPG (A0A068U944)) genes’ activity in *C. arabica* L. for their consistent up-regulation of 1.22-Log2. Those genes are directly involved in trehalose biosynthesis, and trehalose accumulation in plants may improve abiotic-stress tolerance (Figure 8, Figure 9, Figure 10 and Figure 11).

All mentioned TPPs interact with SUS6 (A0A068TQS5) (sucrose synthase 6). SUS6 is actively involved in callose synthesis at the site of phloem sieve elements, and in tuber biomass [43,44]. Furthermore, SUS6 interacts with T22P22.110 (A0A068UW10, a glycosyl hydrolase family 31 protein member), and in turn T22P22.110 interacts with PHS1-3 (A0A068U3V8, an alpha-glucan phosphorylase 1). Phosphorylases are important allosteric enzymatic regulators in carbohydrate metabolism, and in cellular osmotic regulation, mutants can cause embryo arrest (Figure 8) [45,46].

PHS1-3 interacts with ENO2 (A0A068V643) and ENO3 (A0A068ULH0), which are enolase proteins. Enolases are involved in carbon metabolism; they act as a positive regulator of cold stress; they are directly involved in senescence, reproductive, vegetative, and vascular processes; and they are embryo-lethal [47,48]. ENO2 interacts with F16L1.10 (A0A068TYM2, a phosphoglycerate mutase family protein member). F16L1.10 interacts with *PGDH1* (*A0A068TM02*, (3-phosphoglycerate dehydrogenase 1 chloroplastic isoform). It is involved in the plastidial phosphorylated pathway of serine biosynthesis (PPSB) that is required for mature pollen development (Figure 8).

### 3.2. Interaction with Cell Cycle

It is remarkable that proteins that have central roles in the carbon metabolism have established an intricate and tight interaction network in SE-RD29-antTAS somatic embryogenesis. Forty-five proteins related to the cell cycle are induced by stress, and three are induced by water deprivation. This PPI interaction network has a last member: PGDH1. The protein PGDH1 interacts with THY-1 (A0A068UIN3, a bifunctional dihydrofolate reductase-thymidylate synthase 1). It is a key enzyme in folate metabolism playing two different roles: de novo synthesis of tetrahydrofolate or recycling of the dihydrofolate released, depending on the source of dihydrofolate. THY-1 interacts with PCNA2 (A0A068UNF1, proliferating cell nuclear antigen 2). This protein is directly involved in the control of eukaryotic DNA replication (Figure 8 and Figure 9).

### 3.3. PCNA2 Interacts with the Somatic Embryogenesis Gene Module

PCNA2 interacts with NF-YC13 (A0A068V0C8). NF-Y transcription factors play crucial roles in embryogenesis, seed maturation, and SE induction [49,50,51]. Another NF-Y transcription factor directly involved in embryogenesis is *NF-YB9*, which was identified as *LEAFY COTYLEDON1 LEC1* [52,53,54]. NF-YC13 interacts with NF-YA5 (A0A068TPA3), which in turn interacts with NF-YB6 (A0A068U7K3) and NF-YA2 (A0A068UPW6). Those transcriptions factors also interact with NF-YB9/LEC1 (A0A068UXD0). NFYA5 (A0A068TPA3) is involved in the blue light and abscisic acid (ABA) signaling pathways. NF-YB9/LEC1 interacts with NFYA8. NFYA8 (A0A068VIL0 nuclear transcription factor Y subunit A-8) is a transcription factor directly involved in embryo development [55] (Figure 8 and Figure 9). NF-YB9/LEC1 (A0A068UXD0) interacts with AGL15 (A0A068V010, an agamous-like MADS-box protein). AGL15 is a transcription factor involved in the negative regulation of flowering. It prevents premature perianth senescence and abscission, fruit development, and seed desiccation; induces the expression of *DTA4*, *LEC2*, *FUS3*, *ABI3*, *AT4G38680/CSP2*, and *GRP2B/CSP4*; promotes somatic embryo development; and stimulates SE reprogramming via histone acetylation-related mechanisms [56,57] (Figure 8 and Figure 9). AGL15 interacts with VAL2. VAL2 is a transcriptional repressor of the sugar-inducible genes. It is also involved in seed maturation; regulates the expression of *LEC1*, *ABI3*, and *FUS3*, in turn directly impacting embryonic pathways; and regulates the transition from seed maturation to seedling growth, SE, and germination [58,59,60] (Figure 8 and Figure 9). VAL2 interacts with HTR2, a Histone H3.2, a core component of the nucleosome. In coffee *C. arabica* L. cv. Catuaí Amarelo IAC 62, it was found that the CaABI3 activity correlates with the embryogenic potential that is highly expressed in embryogenic masses, and expression of the *VAL2* gene is increased at the end of the embryogenic process [61]. Moreover, AGL15 interacts with BBM. BBM is an AP2-like ethylene-responsive transcription factor. It regulates the expression of *LEC1*, *LEC2*, *FUS3*, and *ABI3*; and it promotes cell proliferation, cellular differentiation, morphogenesis, embryogenesis, and somatic embryogenesis induction [16,62]. And AGL15 also interacts with SERK1 (A0A068TXX7, a somatic embryogenesis receptor kinase 1) (Figure 8 and Figure 9). SERK1 regulates cell proliferation and embryogenic competence, is a central regulator of gametophyte production, regulates the brassinosteroid signaling pathway, and is highly expressed during early embryogenesis stages [63,64,65].

WOX2, AGL15, and NF-YB9/LEC1, all together, interact with FUS3 (A0A068V7Y1, a transcription regulator). FUS3 regulates late embryogenesis and embryo development, controls foliar organ identity, positively regulates the abscisic acid (ABA) synthesis, and negatively regulates gibberellin production. It is also a positive regulator of *ABI3* expression and its protein accumulation in the seed, actively regulates developmental phase transitions and lateral organ development, and is an active regulator during germination (Figure 8 and Figure 9). Mutations in *LEC1* and *FUS3* genes caused embryo lethality due to the loss of desiccation tolerance during late seed development [66,67].

LEC1/NFYB9, NFYB6, and AGL15 interact with ABI3 (A0A068U8A0, a B3 domain-containing transcription factor; Figure 8 and Figure 9). ABI3 participates in abscisic acid-regulated gene expression during seed development and embryo development. It is involved in leaf and embryo degreening, and it regulates the transition between embryo maturation and early seedling development, rather than simply acting as a transducer of the abscisic acid signal [68,69]. ABI3 interacts with AUX1 (A0A068UD59, an auxin transporter protein 1), GEA6 (is an Em-like protein), and GA3OX2. GEA6 is stress-induced, and it is also involved in ABA response, which is required for normal seed development and seed maturation processes. GEA6 interacts with LEA46 (a late embryogenesis abundant protein 46). LEA46 is involved in dehydration tolerance and in the adaptive response to water deficit; it is also involved in somatic embryogenesis [70,71]. AUX1 regulates auxin delivery from the mature phloem to the root meristem via the protophloem cell files. AUX1 interacts with ARR4 (A0A068TWC5, a two-component response regulator (RR); Figure 8 and Figure 9). ARR4 actively participates in the phosphorelay signal transduction system modulating the red light signaling, and it is directly involved in embryogenesis through CK signaling and SAM establishment during maturation of SE [72,73]. ARR4 interacts with AHK5 (A0A068V1M6, histidine kinase 5). AHK5 transmits the stress signal through the MAPK signaling cascade, and it is a negative regulator of the ABA and ethylene signaling pathway inhibiting root elongation and regulates stomatal activity. GA3OX2 (gibberellin 3-beta-dioxygenase 2) participates actively in the gibberellin synthesis, regulates vegetative growth and development, and is an active regulator of embryogenesis [74,75].

SERK1 interacts with WOX2 (A0A068UL49), RLK5 (A0A068TVW5, receptor-like protein kinase 5), and CDC48 (A0A068UGL5, cell division-control protein 48 homolog; Figure 8 and Figure 9). *WOX2* is involved in embryonic development and patterning, and it is highly expressed early during somatic embryo development [76]. CDC48 regulates cell division, development, and growth processes, and it is actively involved in seedling, pollen, and embryo development; mutants are seedling-lethal [77].

CDC48 interacts with HSP70-4 (A0A068UKG5 heat-shock 70 kDa protein 4; Figure 8 and Figure 9).

The heat-shock proteins (HSPs) accumulate in plants in response to high-temperature stress and are expressed as a part of the developmental program of seed maturation of several angiosperms to protect cellular components during seed desiccation [78].

HSP70-4 interacts with TPR repeat-containing thioredoxin TDX, and with HSP70-2 heat-shock 70 kDa protein 2. In Arabidopsis *cpHsc70-1/cpHsc70-2*, the plastid stromal double-mutant is lethal, interferes with ovule development, and reduces pollen-transmission efficiency [79].

It is worth noting the relevant functions carried out by HSP proteins, such as appropriate protein folding and translocation assistance for protein precursors into cellular organelles. They are also involved in leaf and siliques differentiation and proper development, seed maturation processes, flowering, regulation of cytokinins, brassinosteroids, ABA signaling, and regulating plant cell overall transcriptional activity [78,79,80,81,82].

### 3.4. Proline Biosynthesis Module

The SUS6 gene interacts with PAL1 (A0A068VM15, phenylalanine ammonia-lyase 1; Figure 8 and Figure 9). PAL1 is a key enzyme of proline metabolism catalyzing the first reaction in the biosynthesis from L-phenylalanine of a wide variety of natural products based on the phenylpropane skeleton. Also is involved directly in the Salicylic acid biosynthesis and in the response against microbial pathogens. PAL1 (A0A068VM15) interacts with ARGAH1 (A0A068TNZ, Arginase 1, mitochondrial). ARGAH1 catalyzes the hydrolysis of L-arginine to urea and L-ornithine and regulates the urea cycle and the proline and polyamines synthesis. ARGAH1 (A0A068TNZ) interacts with P5CSB (A0A068TXS1, Delta-1-pyrroline-5-carboxylate synthase B) (Figure 8 and Figure 9). P5CSB plays a key role in proline synthesis, is directly involved in the osmoregulation process in plants, embryo development and floral transition [83]. The proline has a central role for plant cell-wall composition, signal transduction cascades, plant development, stem elongation, root and shoot growth, inflorescence architecture, seed development and germination, stress tolerance, modulates the cyclin genes expression, embryo formation, and gametophyte development [84,85,86].

### 3.5. Down-Regulated Genes in SE-RD29-antTAS

The down-regulation of the canonical auxin genes directly implicated during SE induction was found in the RD29-antTAS lines. Fifty-nine genes are related to hormones, of which sixteen respond to auxin; thirteen are auxin-related, interacting with ethylene; six are involved in auxin biosynthesis; two are auxin-responsive to YABB1-5; three are auxin- responsive to SAUR genes; eleven are peroxidases involved in auxin catabolism; two are auxin transporters; and four are WUSCHEL-RELATED HOMEOBOXs (WOX5, WOX11, WOX12, and WOX13) induced by auxins, TCTP1 (translationally controlled tumor protein 1), several peroxidases, glutathione S-transferases, APXs, malate dehydrogenases, thioredoxins, and light-signaling regulators. Furthermore, we found five proteins related to cytokinin regulation, five related to ethylene biosynthesis, fourteen related to jasmonic acid regulation, and eight to gibberellin regulation. This is in accordance with the fact that different transcription factors work in a coordinated manner to orchestrate the stem cell differentiation and maintenance for meristem development.

In summary, silencing the *Trehalase* gene in somatic embryos of coffee (*C. arabica* L.) enabled the generation of osmotic stress-tolerant lines (RD29-antTAS) with enhanced resistance to water deprivation. These RD29-antTAS lines, selected in mannitol-and-sorbitol-containing media, are non-transgenic, since only the endogenous *Trehalase* gene was down-regulated by gene silencing; thus, they are considered to be marker-free and/or intragenic. For several crop species resulting in millions of plants from SE-derived plants like coffee, papaya, cacao, avocado, common-bean, etc., our strategy represents an alternative methodology to overcome the low embryo-to-plant conversion numbers. This marker-free approach may also accelerate the production of resilient coffee varieties, helping to mitigate the negative impacts of climate change and water scarcity on coffee cultivation. The higher competence in SE developed for the SE-RD29-antTAS line might be supported in the regulation model that we propose and is shown in the Figure 15. The *C. arabica* L. antisense *Trehalase* sequence triggers two related processes, trehalose accumulation and trehalose biosynthesis, which in turn modulate carbon metabolism, oxidative stress, and the cell cycle by interacting with the set of genes involved in somatic embryogenesis regulation and establishment. SE-RD29-antTAS-regenerated plants are presently under physiological, biochemical, and molecular analysis to evaluate their adaptability under stress conditions.

## 4. Materials and Methods

### 4.1. Generation of Marker-Free SE-RD29-antTAS of Coffee C. arabica L. var Typica

Genetically modified SE-RD29-antTAS was developed using SE lines of coffee *C. arabica* L. var. Typica [87,88]. The cassette of 4.2 kb of the RD29-antTAS containing the antisense *Trehalase* gene under the RD29 promoter and the NOS terminator was purified and precipitated onto gold particles (1 μm). Prior to bombardment, DNA-coated microprojectiles were resuspended in 400 μL of absolute ethanol. Aliquots of 10 μL (1.66 μg DNA associated with 125 μg of gold particles) were delivered to each macrocarrier membrane; air-dried to remove ethanol; and bombarded onto somatic embryos at the globular stage, using the helium-driven version of PDS-1000/Helium device. The gap distance between the rupture membrane and the flying disk was 1.2 cm, and the macrocarrier (a Kapton disc) traveled 1.2 cm before impact with a steel-stopping screen. The bombardment chamber was evacuated to 0.07 atmospheres, and the gas-acceleration tube was pressurized with the chosen helium gas pressure. Target tissues were placed 7.0 cm from the launch point and bombarded once at 900 PSI. Selection of osmotic stress-tolerant SE lines was performed in the CP2 medium [34], supplemented with mannitol (0.3 M) and sorbitol (0.3 M) (1602 MPa), and 3.0 g/L gelrite pH 8. Twenty embryogenic masses (1 cm^2^, 50 mg fresh weight) containing globular-to-early-torpedo-stage embryos were placed in the center of plates for bombardment. Bombarded SE plates were incubated at 25 ± 2 °C, under a 12/12 h photoperiod at 50 µmol/m^2^-s irradiance, provided by fluorescent lamps T8 Phillips P32T8/TL850 combined with natural light, increasing red/far-red light in the spectrum. Every two weeks, three subcultures were applied to fresh medium in order to obtain resistant SE lines; meanwhile, non-bombarded SE lines stopped growing and developing, and they became necrotic.

### 4.2. Histological Analysis of SEs

Randomly chosen SEs derived from WT-SE and RD29-antTAS lines were collected and fixed in 1 mL of FAE (3.7% formaldehyde, 10% acetic acid glacial, and 50% ethanol) for two hours, at room temperature, followed by dehydration in a series of ethanol dilutions. Fixed and dehydrated samples were embedded in Technovit 7100—Heraeus Kulzer GmbH, Hanau, Hesse (Hessen), Germany plus ethanol (1:1, *v*/*v*) and then embedded in the infiltration solution (Technovit 7100 plus hardener), according to the manufacturer’s instructions [34]. Sections of 10 μm were prepared using a microtome as previously described. Pictures were taken and analyzed on an upright (Leica Microsystems, Wetzlar, Germany).

### 4.3. RNA Isolation and qPCR Analysis

SE-RD29-antTAS and control SE-WT, in globular–early torpedo stage, were used to isolate RNA using TRIzol (Invitrogen, Carlsbad, CA, USA). RNA concentration was measured by its absorbance at 260 nm. The ratio 260 nm/280 nm was assessed, and its integrity was verified by electrophoresis in agarose 2% (*w*/*v*) gels. Samples of cDNA for validations were amplified by PCR using SYBR Green qPCR (Bio-Rad, Hercules, CA, USA) in Real-Time PCR Systems (CFX Bio-Rad, Hercules, CA, USA). The expression of actin, RP29, and S24 was used as reference for calculating the relative amount of target gene expression using the 2^−ΔΔ^*^C^*^t^ method [89]. qPCR analysis was based on at least three biological replicates for each sample with three technical replicates. Oligonucleotides were designed to qPCR (Appendix A [90]). In parallel, the sequencing of cDNA was made in GENEWIZ, Plainfield, NJ, USA. For sequencing, the Illumina HiSeq 2500 (Illumina, San Diego, CA, USA) was applied. A strand-specific protocol was used for RNA-seq library preparation to preserve transcript strand orientation. Raw reads were quality-checked using FastQC and adapters and low-quality bases were trimmed using Trimmomatics [91].

### 4.4. Aligned and Analysis of DEG and GO in the SE-RD29-antTAS Line of Coffee C. arabica L.

Adapter removal was performed using the Trimmomatic v0.3.6 program [91]. RNA-seq reads were aligned RNA-seq reads to the coffee *C. arabica* L. reference genome available in Phytozome v12.1.(https://phytozome.jgi.doe.gov/pz/portal.html, accessed on 11 September 2024) with the STAR aligner v.2.5.2b [92]. In this step, the BAM (Binary Alignment/Map) files were generated. Subsequently, a count and set of transcripts were made using the feature Counts program of the Subread v.1.5.2 package [93]. A quantification and differential analysis of the transcripts was performed using the DESeq2 v1.12.4 program. Finally, an ontology analysis was performed using Blast2GO v.5.2.

Transcriptomic data variability among samples was explored using principal component analysis (PCA). PCA was performed using the prcomp function in R (version 4.3.1). PCA plots were generated using the ggplot2 package (version 3.4.2).

Additionally, statistical differences among groups were evaluated using one-way ANOVA followed by Tukey’s post hoc test, with a significance level set at *p* < 0.05.

### 4.5. PPI Analysis of Up-Regulated Genes Found in SE-RD29-antTAS Line

A gene network with high confidence (0.700) was performed with the STRING database v12 [94], based on *C. canephora*, and homologous in the *Arabidopsis thaliana* genome present in Phytozome, NCBI. The gene identifier (Id) was made according to the UNIPROT [95] and NCBI [96] databases. Homology greater than 60% in *C. canephora* in protein sequence with *A. thaliana* was considered significant.

## 5. Conclusions

Silencing of the *Trehalase* gene in somatic embryos of coffee *C. arabica* L. allowed us to obtain osmotic stress-tolerant RD29-antTAS lines with enhanced resistance to water deprivation.

The RD29-antTAS lines selected in mannitol-and-sorbitol-containing media are not transgenic, since only the *Trehalase* gene of coffee was down-regulated by silencing, and therefore it is considered to be marker-free and/or intragenic.

The transcriptomic analysis revealed the differential expression of 1549 up-regulated genes. A PPI network made by STRING v12.0 with high confidence (0.700) revealed that 230 interact tightly in 10 functional modules.

The SE module consists of 23 genes and fulfills the requirements to develop somatic embryos. This module interacts with the cell cycle, the trehalose biosynthesis, carbon metabolism, oxidative stress, and secondary metabolites to provide osmotic-stress tolerance.

## Figures and Tables

**Figure 1 ijms-26-09224-f001:**
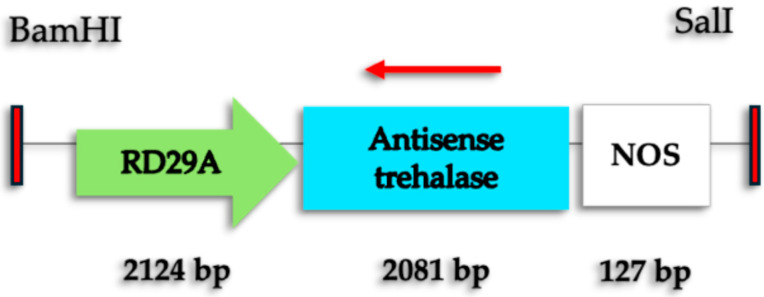
TAS-antisense expression cassette for *C. arabica* L. genetic modification. The expression cassette includes the antisense *C. arabica* L. TAS gene (2081 bp) under the control of the RD29 promoter (2124 bp) and NOS terminator (127 bp). The red arrow indicates that the TAS gene sequence is in antisense orientation.

**Figure 2 ijms-26-09224-f002:**
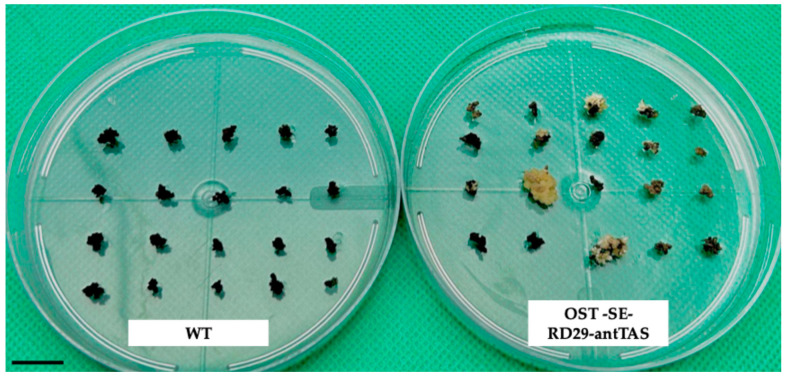
Selection of RD29-antTAS lines of *C. arabica* L. after 2 months of culture in mannitol–sorbitol-containing medium. Notice that WT SE lines were unable to grow in mannitol–sorbitol medium, while bombarded SEs grow and developed. Bar represents 10 mm.

**Figure 3 ijms-26-09224-f003:**
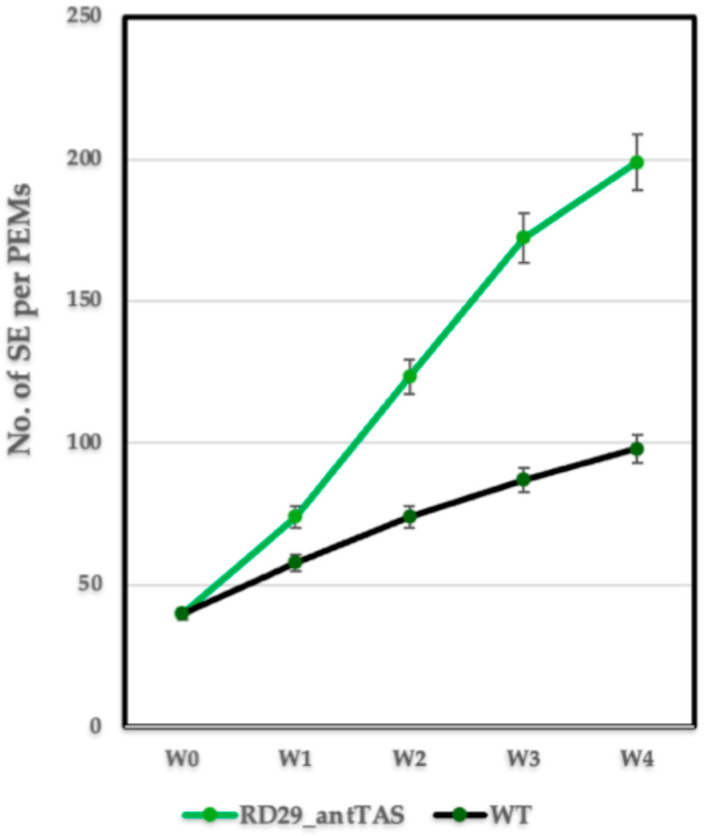
Quantification of SEs derived from one RD29-antTAS line compared to the WT-SE line.

**Figure 4 ijms-26-09224-f004:**
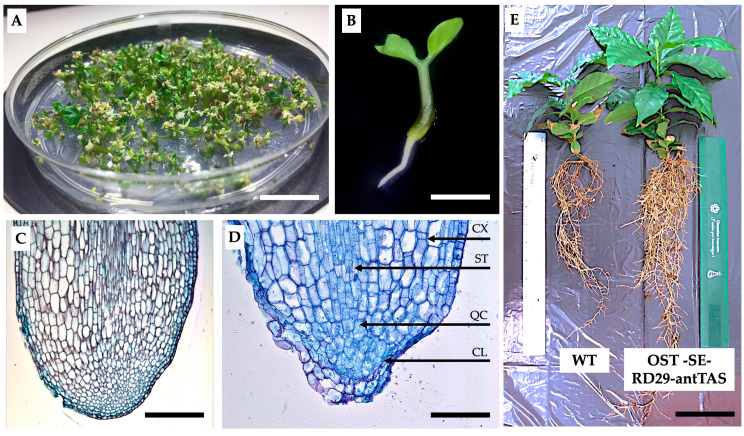
SEs’ maturation and plant regeneration of RD29-antTAS lines of *C. arabica* L. (**A**) SEs of *C. arabica* var. Typica at torpedo stage containing 200 SEs in the maturation medium. Bar represents 10 mm. (**B**) Germination of SEs of coffee *C. arabica* L. after one month in culture. Bar represents 2 mm. (**C**) SEs at cotyledonary stage; notice that RAM is not developed. Bar indicates 100 μm. (**D**) SEs at cotyledonary stage; notice that RAM is well developed. SAM, shoot apical meristem; LP, leaf primordium; RAM, root apical meristem; CL, columella; ST, stella; CX, cortex; QC, quiescent center. Bar indicates 100 μm. (**E**) Morphology of a regenerated plant from WT-SE (left) and one derived from RD29-antTAS lines (left) after eight months in soil conditions under a growth chamber. Plants derived from osmotic-stress RD29-antTAS are robustness, with a higher leaf and root area than plants from non-osmotic medium. Bar represents 10 mm.

**Figure 5 ijms-26-09224-f005:**
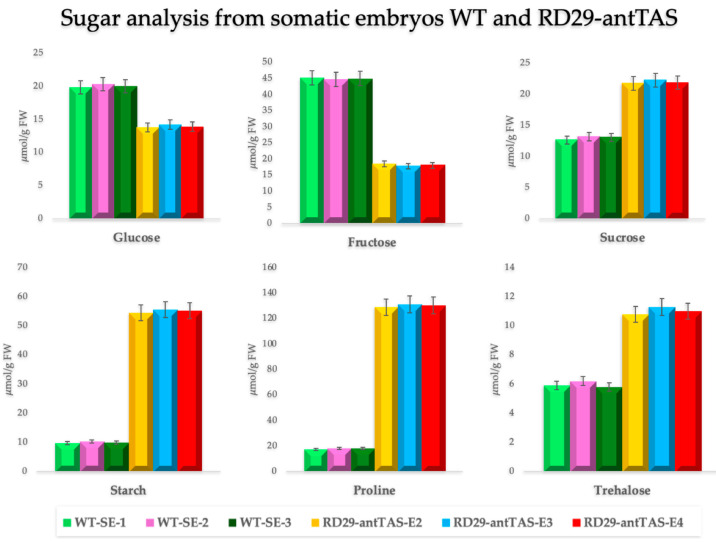
Levels of soluble sugars (glucose, fructose, sucrose, and trehalose), starch, and proline in WT-SE and SE-RD29-antTAS lines after 30 days in MS medium with 0.3 M mannitol + 0.3 M sorbitol (−1.602 MPa). SE-RD29-antTAS lines showed reduced glucose (~32%) and fructose (~60%), and increased sucrose (~69%), starch (5.5-fold), proline (7.6-fold), and trehalose (1.8-fold), indicating enhanced accumulation of osmoprotectants. Data are mean ± SD. Different letters indicate significant differences (*p* < 0.05).

**Figure 6 ijms-26-09224-f006:**
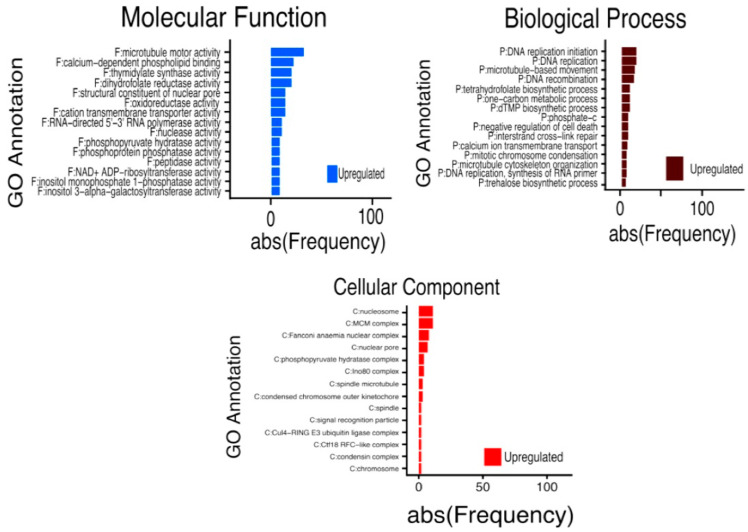
Exclusive molecular functions, biological process, and cellular components regulated during SE-RD29-antTAS of coffee *C. arabica* L. cv Typica under osmotic stress.

**Figure 7 ijms-26-09224-f007:**
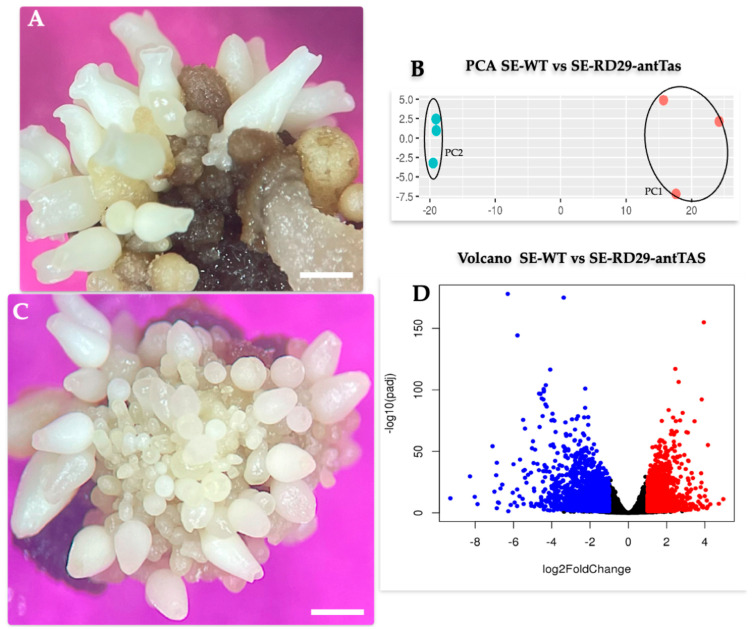
Somatic embryogenic lines of RD29-antTAS and SE-WT; about 0.5 cm^2^ was used for transcriptome analysis. (**A**) SE-WT, Bar represents 1.5 mm. (**B**) Principal component analysis of the SE-lines: PC1, WT-SE, 92% of variance; PC2, RD29-antTAS, 4% of variance; (**C**) RD29-antTAS. Bar represents 1.5 mm. (**D**) Volcano plot showing up- and down-regulated genes in the embryogenic lines WT-SE vs. RD29-antTAS. Each experiment was repeated three times. Tukey’s test was used to detect significant difference at *p* ≤ 0.05.

**Figure 8 ijms-26-09224-f008:**
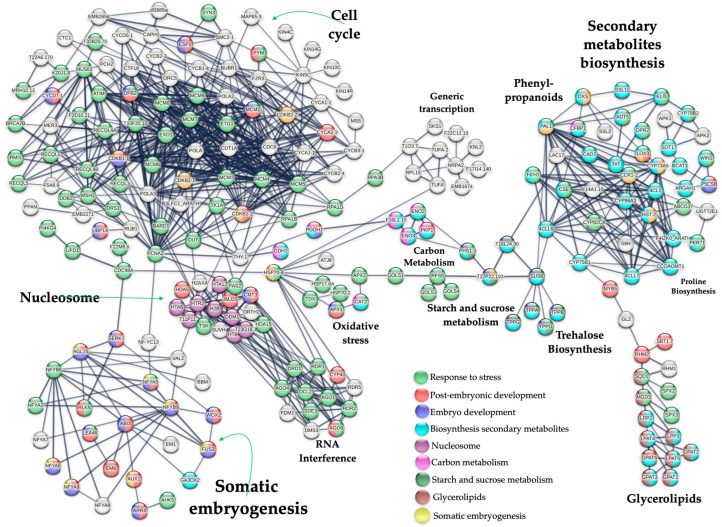
PPI network of up-regulated genes derived from the STRING database v12.0 of coffee *C. arabica* L. from the transcriptomic-wide analysis with high confidence (0.700). Modules are highlighted with the name of the function. The figure represents a full network; the edges indicate both functional and physical protein associations.

**Figure 9 ijms-26-09224-f009:**
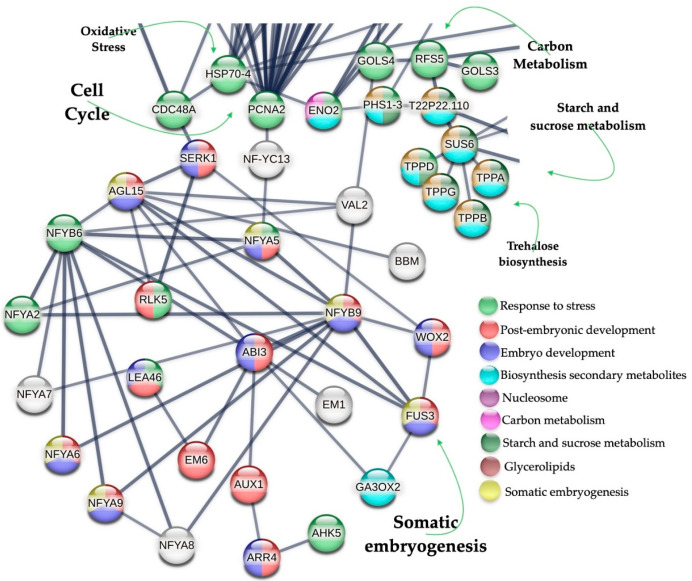
PPI network of up-regulated genes involved in SE derived from the STRING database v12.0 of coffee *C. arabica* L. with high confidence (0.700). The color of spheres represents the name of the function. The figure represents a full network; the edges indicate both functional and physical protein associations.

**Figure 10 ijms-26-09224-f010:**
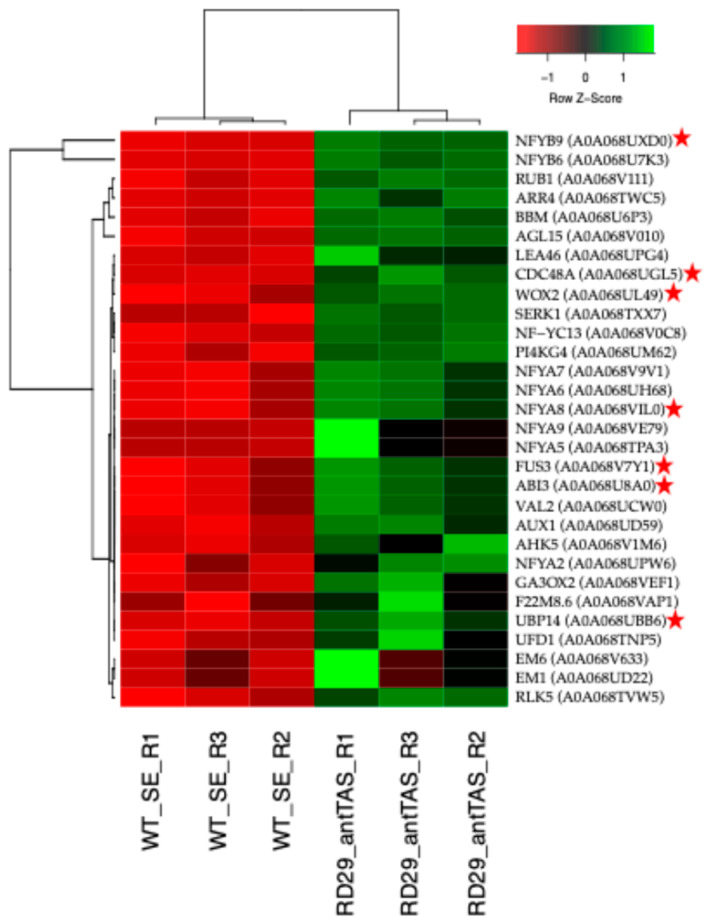
Hierarchical clustering analyses (HCAs) of up-regulated genes involved in the SE module derived from SE-RD29-antTAS under osmotic medium. Genes labeled (*) in red are embryo-lethal. Levels of up-regulation are shown in Log2. Row-wise Z-score normalization. This standardization highlights expression patterns across conditions but does not reflect absolute Log_2_FC magnitude.

**Figure 11 ijms-26-09224-f011:**
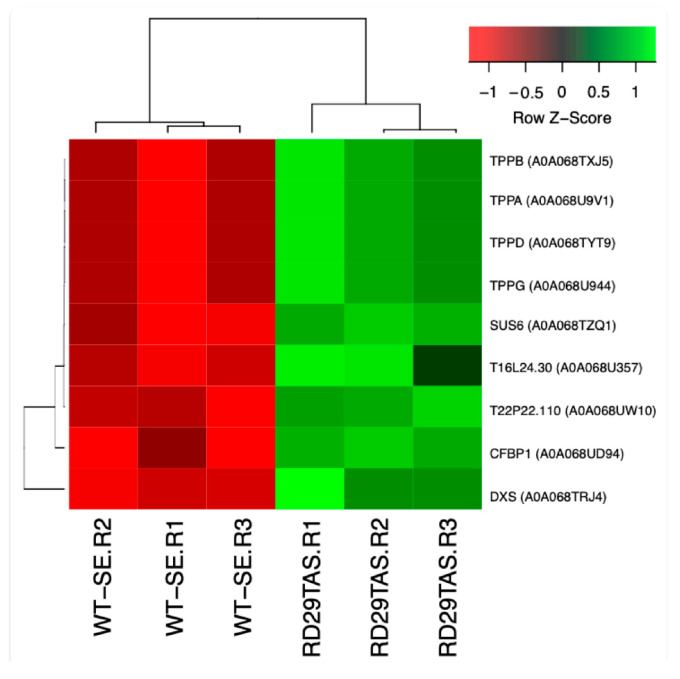
Hierarchical clustering analyses (HCAs) of up-regulated genes involved in the trehalose biosynthesis module in SE-RD29-antTAS development under osmotic medium. Levels of up-regulation are shown in Log2. The original expression data represent Log_2_ fold change (Log_2_FC) values relative to the SE-WT control. To enhance visualization of expression patterns across genes, data were normalized using row-wise Z-score transformation.

**Figure 12 ijms-26-09224-f012:**
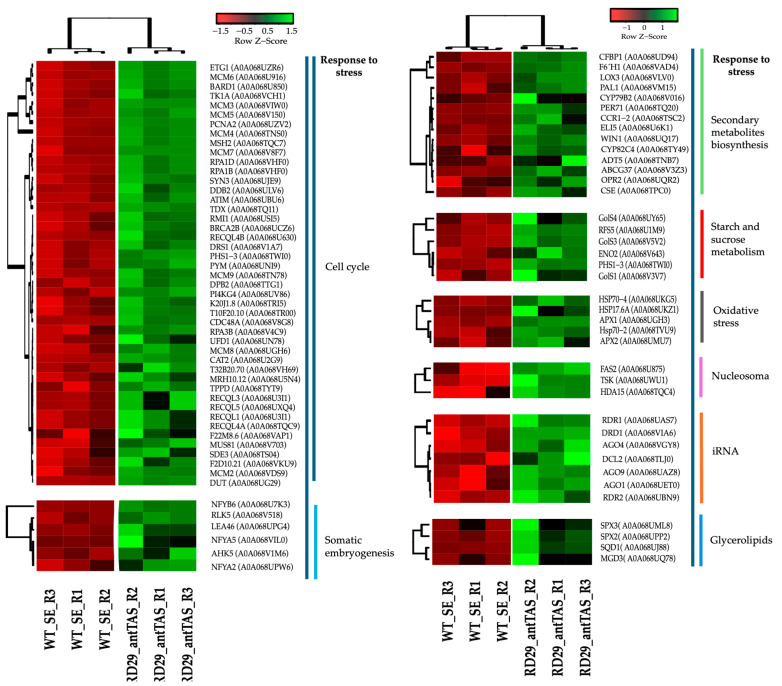
Hierarchical clustering analyses (HCAs) of up-regulated genes involved in response to stress in SE-RD29-antTAS development under osmotic medium. Levels of up-regulation are shown in Log2. The original expression data represent Log_2_ fold change (Log_2_FC) values relative to the SE-WT control. To enhance visualization of expression patterns across genes, data were normalized using row-wise Z-score transformation.

**Figure 13 ijms-26-09224-f013:**
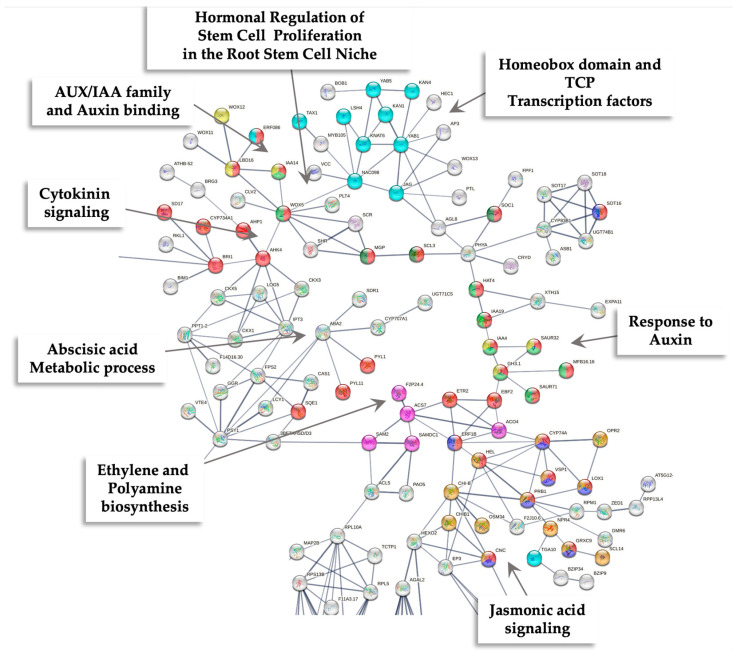
PPI network of down-regulated genes derived from the STRING database v12.0 of coffee *C. arabica* L. from the transcriptomic-wide analysis with 0.600 confidence. Modules are highlighted with the name of the function. The figure represents a full network; the edges indicate both functional and physical protein associations.

**Figure 14 ijms-26-09224-f014:**
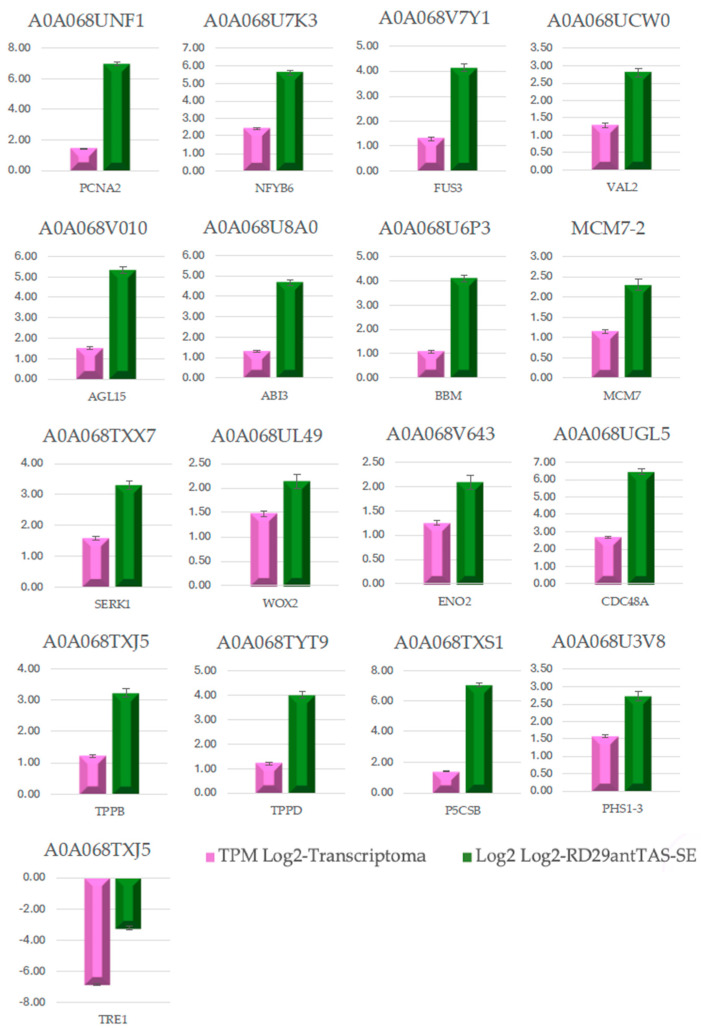
Validation of the transcriptome-wide analysis by quantitative reverse transcription PCR (qRT-PCR) of 16 up-regulated DEGs involved in somatic embryogenesis, trehalose biosynthesis, carbon metabolism, and the cell cycle, as well as the down-regulated *Trehalase* gene. Values are expressed in Log2.

**Figure 15 ijms-26-09224-f015:**
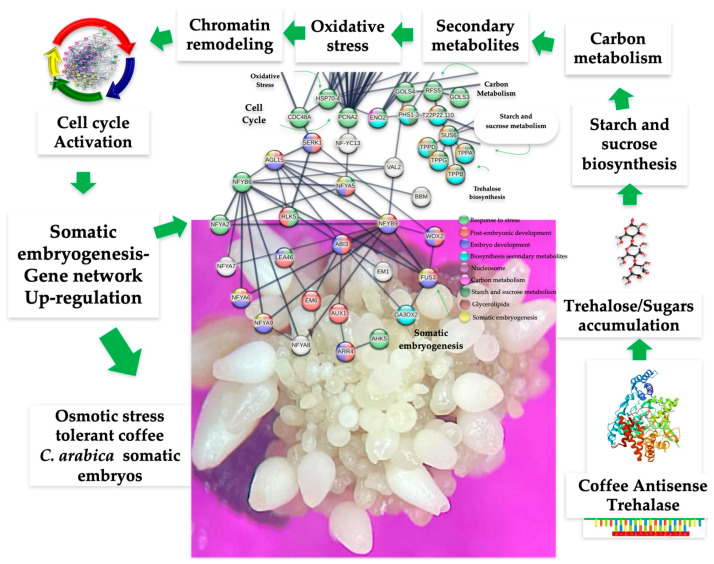
Model of the transcriptomic analysis of osmotic stress-tolerant somatic embryos of *Coffea arabica* L. mediated by coffee antisense *Trehalase* gene: a marker-free approach.

**Table 1 ijms-26-09224-t001:** Summary of genetic modification events of *C. arabica* L. mediated by particle bombardment of somatic embryos with RD29-antTAS under different osmotic conditions.

Treatment	Bombarded Plates	RD29-antTAS Lines	RD29-antTAS Lines/Plate	Plant Conversion (%)
**Mannitol 0.15 M + Sorbitol 0.15 M**	30	33	1.1	25%
**Mannitol 0.3 M + Sorbitol 0.3 M**	30	85	2.8	90%
**Mannitol 0.45 M + Sorbitol 0.45 M**	30	6	0.4	15%

## Data Availability

The transcriptome data can be found in the Bioproject PRJNA 1067141. The link is: https://www.ncbi.nlm.nih.gov/bioproject/?term=PRJNA1067141 (accessed on 22 November 2024).

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
