# Peer review of "Transcriptomic Analysis of Osmotic Stress-Tolerant Somatic Embryos of Coffea arabica L. Mediated by the Coffee Antisense Trehalase Gene: A Marker-Free Approach"

_ijms, 2025, doi:10.3390/ijms26189224_

Round 1
Reviewer 1 Report
Comments and Suggestions for Authors
In the manuscript of Transcriptomic analysis of osmotic stress tolerant somatic embryos of Coffea arabica L. mediated by the coffee antisense Trehalase gene: A Marker free approach, the authors perform a lot of works to identify their conclusion. The experimental layout is consequential, the methods used is appropriate, and the techniques are well performed and in accordance with their purpose. Most of the conclusions are supported by the data presented.
I have some comments and would recommend the work for publication after revision.
Comments:
- The differential gene screening threshold (|log2FC|≥2) was too lenient and did not account for FDR correction values (e.g., padj<0.05). It is recommended to supplement the significance threshold line for volcano maps.
- Section 4.1 does not describe bombardment parameters (gold powder particle size, bombardment distance) Section 4.3 RNA-seq library construction type (strand-specific/non-stranded) is missing.
- Embryo sections in Figure 4C/D are not labeled with developmental stages (e.g., cardioid phase/torpedo phase ratio). Suggested arrows indicate key structural differences (e.g., QC regions) and supplement scale values.
- The STRING analysis in Figure 8 does not show the key parameters: the node size is not associated with the degree of connectivity, and the enrichment p-value is not marked. It is recommended to supplement the co-expression heat map of the core genes in the module.
- Please add the research that already exists on stress-response genes, there is only one argument in the discussion, which is too little. The interaction between the embryogenesis module (WOX2/SERK1/NF-Y) and stress-response genes (e.g., HSP70) shown in Figure 9 is not explored in depth.
Author Response
Review 1:
- The differential gene screening threshold (|log2FC|≥2) was too lenient and did not account for FDR correction values (e.g., padj<0.05). It is recommended to supplement the significance threshold line for volcano maps.
- Section 4.1 does not describe bombardment parameters (gold powder particle size, bombardment distance)
We appreciate the reviewer’s observation. In section 4.1 material and methods, the reference [86], that corresponds to Valencia-Lozano (2021) protocol, Valencia-Lozano, E.; Cabrera-Ponce, J.L.; Noa-Carrazana, J.C.; & Ibarra, J.E. Coffea arabica L. resistant to coffee berry borer Hypothenemus hampei mediated by expression of the Bacillus thuringiensis Cry10Aa protein. Front. Plant Sci. 2021, 12, 765292. DOI: https://doi.org/10.3389/fpls.2021.765292, details the Biobalistic procedure, was cited as requested. Nevertheless, following the reviewer’s suggestion, we have now included specific bombardment parameters—such as the gold particle size and bombardment distance—directly in Section 4.1 to provide a clearer and more comprehensive description within the manuscript.
Previously it was in the Line 578, modified is in lane 608. Modifications according to reviewer 1 are marked in green color.
4.1. Generation of marker-free SE-RD29-antTAS of coffee C. arabica L. var Typica.
Genetically modified SE-RD29-antTAS was developed using SE lines of coffee C. arabica L. var. Typica [86]. Cassette RD29-antTAS containing the antisense Trehalase gene under the RD29 promoter and the NOS terminator, it was purified 1.5 Kb and precipitated onto gold particles (1 μm); prior to bombardment, DNA-coated microprojectiles were resuspended in 400 μL of absolute ethanol. Aliquots of 10 μL (1.66 μg DNA associated with 125 μg of gold particles) were delivered to each macrocarrier membrane, air-dried to remove ethanol, and bombarded onto somatic embryos at the globular stage, using the helium-driven version of PDS-1000. The gap distance between the rupture membrane and the flying disk was 1.2 cm, the macrocarrier (a kapton disk) traveled 1.2 cm before impact with a steel-stopping screen. The bombardment chamber was evacuated to 0.07 atmospheres, and the gas acceleration tube was pressurized with the chosen helium gas pressure. Target tissues were placed 7.0 cm from the launch point and bombarded once at 900 psi. Selection of osmotic-stress tolerant SE lines were done in the CP2 medium [34], supplemented with mannitol (0.3M) and sorbitol (0.3M) (1,602 MPa) and 3.0 g/L gelrite pH 8. Twenty embryogenic masses (1 cm2, 50 mg fresh weight) containing globular to early torpedo stage were placed in the center of plates for bombardment.. Bombarded SE plates were incubated at 25±2°C, under a 12/12 h photoperiod at 50 µmol/m2-s irradiance provided by fluorescent lamps T8 Phillips P32T8/TL850 combined with natural light increasing red/far red light in the spectrum. Three subcultures every two weeks onto fresh medium were applied to get resistant SE lines while non bombarded SE lines deceased growth and development and became necrotic.
- Section 4.3 RNA-seq library construction type (strand-specific/non-stranded) is missing.
We thank the reviewer for pointing out this omission. The RNA-seq libraries used in this study were constructed using a strand-specific protocol, which preserves the orientation of the transcript. This information has now been included in Section 4.3 of the revised manuscript.
Line 600
4.3. RNA isolation and qPCR Analysis
SE-RD29-antTAS and control SE-WT, in globular-early torpedo stage were used to iso-late RNA using Trizol (Invitrogen, Carlsbad, CA, USA). RNA concentration was meas-ured by its absorbance at 260 nm, the ratio 260 nm/280 nm was assessed, and its integ-rity was verified by electrophoresis in agarose 2% (w/v) gels. Samples of cDNA for val-idations were amplified by PCR using SYBR Green qPCR (Bio-Rad, Hercules, CA, USA) in Real-Time PCR Systems (CFX Bio-Rad, Hercules, CA, USA). The expression of actin, RP29, and S24 was used as reference for calculating the relative amount of target gene expression using the 2−ΔΔCt method [87]. qPCR analysis was based on at least three biological replicates for each sample with three technical replicates. Oligonucleotides were designed to qPCR (supplementary Table S24). In parallel the sequencing of cDNA was made in GENEWIZ, Plainfield, NJ, USA. To sequence the Illumina HiSeq 2500 (Illumina, San Diego, CA, USA). A strand-specific protocol was used for RNA-seq library preparation to preserve transcript strand orientation. Raw reads were quality-checked using FastQC (http://www.bioinformatics.babraham.ac.uk/projects/fastqc/) and adapters and low-quality bases were trimmed using Trimmomatics [88].
- Embryo sections in Figure 4C/D are not labeled with developmental stages (e.g., cardioid phase/torpedo phase ratio). Suggested arrows indicate key structural differences (e.g., QC regions) and supplement scale values.
We appreciate the reviewer’s observation. Figure 4C/D has been updated to include labels indicating embryo developmental stages (e.g., heart, torpedo), arrows marking key structural features such as the QC region, and appropriate scale bars with values. These additions improve clarity and anatomical interpretation.
(C) SEs at cotyledonary stage, notice that RAM is not developed. Bar indicates 100 μm. (D) SEs at cotyledonary stage, notice that RAM is well developed. SAM, shoot apical meristem, LP, leaf primordium. Root apical meristem (RAM). CL, columella, ST, stella, CX, cortex, QC, quiescent center. Bar indicates 100 μm.
- The STRING analysis in Figure 8 does not show the key parameters: the node size is not associated with the degree of connectivity, and the enrichment p-value is not marked. It is recommended to supplement the co-expression heat map of the core genes in the module.
Done. Figure 8 footnote was modified according to reviewer’s suggestions.
- Please add the research that already exists on stress-response genes, there is only one argument in the discussion, which is too little. The interaction between the embryogenesis module (WOX2/SERK1/NF-Y) and stress-response genes (e.g., HSP70) shown in Figure 9 is not explored in depth.
Done. References were added, lines 519-520.

Reviewer 2 Report
Comments and Suggestions for Authors
The manuscript describes the investigation where the Trehalase gene silencing was applied to allow trehalose accumulation favoring plant surviving in extreme drought/salt environments, which led to the osmotic stress-tolerant OST-SE lines with enhanced capability of resistance to water deprivation.
The topic and scope fit the journal’s special issue “Molecular Mechanisms of Plant Abiotic Stress Tolerance;” the methods are appropriate, the result dissemination is adequate and supports the Conclusion.
Overall, the manuscript is not well edited with poor English usage that hinders efficient narrative, and inconsistency exists in several places, lacking scientific rigor. Reproducibility may be improved by adding missing experimental details. Significance statement is missing.
Details for consideration during the revision:
- The last 3 paragraphs of Introduction appear more like a summary than Objectives. Please introduce the objectives and hypothesis if there is one, instead.
- Fig. 1 What 2124 bp, 2081 bp, etc. mean should be explained.
- Line 117 condition format is not consistent with line 119, nor Table 1. Numerical results are not consistent with Table 1.
- the bar scales must be marked directly on the graphs, not in captions.
- Fig. 4E, labels are not the same as caption. should be left-right, not left-left.
- Fig. 5, what is the time scale each legend represents is not clear.
- OST-SE is used interchangeably with SE-RD29-antTAS throughout. This should be clarified.
- Line 193-106, there is wrong usage of . in place of , in the 2 numbers-
- Line 198, 3850 and 999 do not add up to 4879.
- A Statistical analysis section introducing Tukey's test and PCA details is missing
- Fig. 10-12, where log2 is plotted is not clear.
- Fig. 14, y-axis in each panel does not have any parameter. Please add it.
- Additional references are needed to strengthen Line 549 “…The proline has a central role for plant cell wall composition… gametophyte development [84, 85, ref].”
Recommended ref = Sahai, MA, et al. "First principle computational study on the full conformational space of l-proline diamides." The Journal of Physical Chemistry A 109 (2005): 2660-2679.
- The practical indications and significance of the current study should be discussed in Discussion and re-stated in the Conclusions.
- The Supplementary Information file should be submitted as a single pdf file containing all supporting information.
Comments on the Quality of English Language
thorough English editing is necessary.
Author Response
Review 2.
The manuscript describes the investigation where the Trehalase gene silencing was applied to allow trehalose accumulation favoring plant surviving in extreme drought/salt environments, which led to the osmotic stress-tolerant OST-SE lines with enhanced capability of resistance to water deprivation.
The topic and scope fit the journal’s special issue “Molecular Mechanisms of Plant Abiotic Stress Tolerance;” the methods are appropriate, the result dissemination is adequate and supports the Conclusion.
Overall, the manuscript is not well edited with poor English usage that hinders efficient narrative, and inconsistency exists in several places, lacking scientific rigor. Reproducibility may be improved by adding missing experimental details. Significance statement is missing.
Details for consideration during the revision:
Details for consideration during the revision:
- The last 3 paragraphs of Introduction appear more like a summary than Objectives. Please introduce the objectives and hypothesis if there is one, instead.
We appreciate the reviewer’s observation. We have modified the introduction by adding our hypothesis. Modifications are marked with red colour.
- Introduction
Coffee is a perennial tropical crop originated in Ethiopia, characterized by abundantly distributed rainfall [1]. Coffee depends on the environment and an increase of a few degrees of average temperature and/or short periods of drought can substantially de-crease yields of quality. This could result in environmental, economic, and social prob-lems [2]. In Brazil drought and frost decreased 25% the yield of coffee beans in 2021 and is expected reductions up to 60% [3]. This has triggered considerable tension on in-ternational markets, leading to a two-fold increase in coffee prices. According to a re-cent study, a 2–2.5 °C temperature increase would considerably reduce the available coffee growing area [4].
The molecular mechanisms that Impact drought in coffee physiology and yield has been reported [5 – 12]. However, the process of identifying and utilizing these traits is lengthy. An alternative tool to overcome these problems is the plant biotechnology by means of genetic transformation and genome editing using the somatic embryogenesis (SE) process. SE occurred when an embryonic stem cell is induced from a somatic cell that differentiates into a somatic embryo (SEs), with the capacity to develop a plant con-taining the same genetic information as its precursor and sets the template for post-embryonic development and sculpt the adult body pattern.
SE requires the transcriptional regulation of a set of genes in response to stress mediat-ed by plant hormones, osmotic, heavy metals, salinity, signaling elements that triggers cellular reprogramming and transformation of somatic cells into somatic embryos. The application of transcriptomic analysis has revealed a great number of differentially ex-pressed genes (DEGs) during SE in several crops and Arabidopsis thaliana [for review see: 13 – 27].
One molecular approach is the role of trehalose (TRE) in plant surviving in extreme drought/salt environments. TRE accumulation in plant improves abiotic stress toler-ance. TRE is a non-reducing disaccharide of glucose (a-D-glucopyran-osyl-a-D-glucopyranoside) that serves as a reserve metabolite in yeast and fungi [28]. TRE stabi-lizes proteins and lipid membranes. TRE is synthesized in a two-reaction process, in which trehalose-6- phosphate (T6P) is first synthesized from glucose-6-P and UDP-glucose by the enzyme trehalose phosphate synthase (TPS) and subsequently de-phosphorylated by trehalose-6P phosphatase (TPP) [29]. In plants, Trehalase (TAS) activi-ty hydrolyses TRE and maintains its concentration at low levels, to prevent detrimental effects. TA is present in all organs of higher plants, with the highest activities in flow-ers [30].
Inhibition of TAS activity by validamicin-A in A. thaliana, led to changes in TRE and sucrose contents in different parts of the plant [30]. In 2004, Gámez-Escobedo [31] in-duced an increase in tobacco plant regeneration in osmotic-stress medium using the al-falfa TAS antisense coupled with the RD29 promoter of A. thaliana. Later, the same cas-sette was used to produce drought tolerant maize B73 inbred line and tested the gener-ation T4 under field conditions successfully [32, 33]. As it occurred in tobacco a higher capability of plant regeneration derived from somatic embryos was observed in the B73 inbred line.
In this study, we aimed to evaluate whether the inhibition of TAS activity in C. arabica increases internal TRE levels, thereby promoting improved embryogenic competence, root meristem development, and overall plant biomass under osmotic stress conditions.
For this purpose, we constructed a gene cassette containing the C. arabica TAS gene in antisense orientation, driven by the C. arabica RD29 promoter and terminated by the NOS terminator, and used it to genetically transform an embryogenic C. arabica cv. Typica line by particle bombardment. The transformed lines were subjected to osmotic stress using medium supplemented with 0.3 M mannitol and 0.3 M sorbitol. conditions. Transcriptomic analyses were performed on high-competence SE lines to elucidate the molecular mechanisms involved in the improved somatic embryo development.
The transcriptome analysis revealed 1,549 up-regulated (Log2 [fold change (FC)] ≥2.0) and 2,301 down-regulated (Log2 [FC] ≤ −2.0) genes. A PPI network mediated by STRING database v12.0 with high confidence (0.700) was performed to understand the molecular mechanisms involved in the SE process.
- 1 What 2124 bp, 2081 bp, etc. mean should be explained.
It was explained
Figure 1. TAS-antisense expression cassette for C. arabica L. genetic modification. The expression cassette includes the antisense C. arabica L. TAS gene (2081 base pairs) un-der the control of the RD29 promoter (2124 base pairs) and NOS terminator (127 base pairs).
- Line 117 condition format is not consistent with line 119, nor Table 1. Numerical results are not consistent with Table 1.
We thank the reviewer for this careful observation. The paragraph has been revised to ensure consistency in both the format of the osmotic treatments and the numerical data reported. Now, all treatment names follow a standardized format (e.g., “mannitol 0.3 M + sorbitol 0.3 M”) as presented in Table 1, and all numerical values related to OST-SE line counts, lines per plate, and plant conversion percentages have been corrected to match those in the table. This adjustment improves the clarity and accuracy of the manuscript.
Line 121-129
Among the osmotic treatments tested, the most effective condition for selection was mannitol 0.3 M + sorbitol 0.3 M, which yielded a total of 85 OST-SE lines, averaging 2.8 lines per bombarded plate, and resulting in a 90% plant conversion rate. In comparison, the medium containing mannitol 0.15 M + sorbitol 0.15 M produced 33 OST-SE lines (1.1 lines/plate) with a 25% conversion rate, while the condition with mannitol 0.45 M + sorbitol 0.45 M generated only 6 lines (0.4 lines/plate) and a 15% conversion rate. These results, summarized in Table 1, indicate that the intermediate osmotic pressure condition (0.3 M of each sugar alcohol) is optimal for selecting osmotic stress-tolerant somatic embryo lines with high regeneration potential.
- the bar scales must be marked directly on the graphs, not in captions. The bar scales are also accepted to be mentioned in the figure footnote, as well as in the photographs inside the figures.
Ok. Done
- 4E, labels are not the same as caption. should be left-right, not left-left.
Dear Reviewer 2 we modified the figure
- 5, what is the time scale each legend represents is not clear.
Levels of soluble sugars (glucose, fructose, sucrose, trehalose), starch, and proline in WT-SE and SE-RD29-antTAS lines after 30 days in MS medium with 0.3 M mannitol + 0.3 M sorbitol (−1.602 MPa). SE-RD29-antTAS lines showed reduced glucose (~32%) and fructose (~60%) and increased sucrose (~69%), starch (5.5-fold), proline (7.6-fold), and trehalose (1.8-fold), indicating enhanced accumulation of osmoprotectants. Data are mean ± SD. Different letters indicate significant differences (p < 0.05).
- OST-SE is used interchangeably with SE-RD29-antTAS throughout. This should be clarified.
Dear reviewer 2 we have unified within the manuscript using only the term: SE-RD29-antTAS
- Line 193-106, there is wrong usage of . in place of , in the 2 numbers-
Dear Reviewer 2, we have made the correction
- Line 198, 3850 and 999 do not add up to 4879.
Dear Reviewer 2, we have made the correction
- A Statistical analysis section introducing Tukey's test and PCA details is missing. Lines 604-605.
Dear Reviewer 2, we have made the correction
- 10-12, where log2 is plotted is not clear.
We thank the reviewer for pointing out the lack of clarity regarding the logâ‚‚ data representation in Figures 10–12. The figure legends have been revised to specify that the values represent logâ‚‚ fold change (logâ‚‚FC) relative to SE-WT. We also clarified whether any additional normalization row Z-score was applied. These adjustments ensure proper interpretation of the data shown.
Line 315
Figure 10. Heatmap of differentially expressed genes showing expression levels as logâ‚‚ fold change (logâ‚‚FC), followed by row-wise Z-score normalization. This standardization highlights expression patterns across conditions but does not reflect absolute logâ‚‚FC magnitude. Genes marked with (*) are embryo lethal.
Line 326
Figure 11. Hierarchical clustering analysis (HCA) of up-regulated genes involved in the trehalose biosynthesis module during SE-RD29-antTAS development under osmotic stress conditions. The original expression data represent logâ‚‚ fold change (logâ‚‚FC) values relative to the SE-WT control. To enhance visualization of expression patterns across genes, data were normalized using row-wise Z-score transformation.
Line 340
Figure 12. Hierarchical clustering analysis (HCA) of up-regulated genes involved in the response to stress during SE-RD29-antTAS development under osmotic stress conditions. The original expression data represent logâ‚‚ fold change (logâ‚‚FC) values relative to the SE-WT control. To enhance visualization of expression patterns across genes, data were normalized using row-wise Z-score transformation.
- 14, y-axis in each panel does not have any parameter. Please add it. Done
- Additional references are needed to strengthen Line 549 “…The proline has a central role for plant cell wall composition… gametophyte development [84, 85, ref].”
- Recommended ref = Sahai, MA, et al. "First principle computational study on the full conformational space of l-proline diamides." The Journal of Physical Chemistry A 109 (2005): 2660-2679. Reference added.
- The practical indications and significance of the current study should be discussed in Discussion and re-stated in the Conclusion
Silencing the Trehalase gene in somatic embryos of coffee (C. arabica L.) enabled the generation of osmotic stress-tolerant lines (RD29-antTAS) with enhanced resistance to water deprivation. These RD29-antTAS lines, selected in mannitol and sorbitol-containing media, are non-transgenic since only the endogenous Trehalase gene was down-regulated by gene silencing, thus considered marker-free and/or intragenic. In several crop species demanding millions of plants from SE derived-plants like coffee, papaya, cacao, avocado, common-bean, etc. our strategy represents and alternative methodology to overcome the low embryo to plant conversion numbers. This marker-free approach may also accelerate the production of resilient coffee varieties, helping to mitigate the negative impacts of climate change and water scarcity on coffee cultivation.
- The Supplementary Information file should be submitted as a single pdf file containing all supporting information.
We appreciate the reviewer’s suggestion. We have consolidated all supplementary materials into a single PDF file containing all supporting information, and the updated file has been submitted accordingly.

Round 2
Reviewer 1 Report
Comments and Suggestions for Authors
accept
Author Response

(The authors gave the same response as above.)

Reviewer 2 Report
Comments and Suggestions for Authors
Half of the issues have been addressed satisfactorily.
The other half, although authors have responded as "Done" or "have made the correction," have not been done or corrected following this reviewer's suggestion.
Specifically, points 8,9,10,12 have not been corrected.
13. the suggested reference for strengthening the statements has not been added.
15. the supplementary file is the same as the previous version. Please upload the updated single pdf file.
Some figures have lower resolution (visual clarity) than the previous version. Please fix this.
Author Response
Review 2:
Comment 1: Line 193-106, there is wrong usage of . in place of , in the 2 numbers.
Response 1: Lines 103–106: There is incorrect usage of a period (.) instead of a comma (,), in two numbers. The corrected sentence should read:
The transcriptome analysis revealed 1,549 up-regulated (logâ‚‚ fold change [FC] ≥ 2.0) and 2,301 down-regulated (logâ‚‚ FC ≤ −2.0) genes. A protein–protein interaction (PPI) network was constructed using the STRING database v12.0 with high confidence (score ≥ 0.700) to better understand the molecular mechanisms involved in the SE process.
Lines 203–206: … This produced a total of 409,947,036 sequence reads, encompassing 122,984 Mbases from all four cDNA libraries, SE-RD29-antTAS and SE-WT. On average 90.6% of the quality filter passed reads generated for all three somatic embryo samples were mapped uniquely to the reference genome and Q-40 mean of 37.9. Reviewer comment:
Comment 2: Line 198, 3,850 and 999 do not add up to 4,879.
Response 2: Thank you for your comment. We have corrected the values to reflect the accurate classification. The total of 4,879 expressed genes includes 3,850 protein-coding genes and 1,029 non-coding genes, the latter category now encompassing other non-protein-coding features such as pseudogenes and unclassified transcripts (999 + 30). The manuscript has been updated accordingly.
Comment 3: A Statistical analysis section introducing Tukey's test and PCA details is missing. Done. Lines 604–605.
Response 3: Thank you for your comment. We have added the Statistical Analysis section in Lines 665–669 as follows:
Transcriptomic data variability among samples was explored using Principal Component Analysis (PCA). PCA was performed using the prcomp function in R (version 4.3.1). PCA plots were generated using the ggplot2 package (version 3.4.2). Additionally, statistical differences among groups were evaluated using one-way ANOVA followed by Tukey’s post-hoc test, with a significance level set at p < 0.05.
Comment 4: Fig. 14, y-axis in each panel does not have any parameter. Please add it.
Response 4: Thank you for your observation. We have modified Figure 14 to include the parameter on the y-axis. Specifically, the y-axis now shows logâ‚‚ values, as appropriate for the data presented.
Comment 5: the suggested reference for strengthening the statements has not been added. Ok reference 86.
Comment 6 : The supplementary file is the same as the previous version. Please upload the updated single PDF file. Some figures have lower resolution (visual clarity) than the previous version. Please fix this.
Response 5: Thank you for your comment. We have corrected the issues as follows:
Supplementary File: The updated supplementary file has now been uploaded as a single PDF document, replacing the previous version.
Figure Resolution: Figures with reduced visual clarity have been replaced with high-resolution versions to ensure optimal readability and presentation quality.

Round 3
Reviewer 2 Report
Comments and Suggestions for Authors
It is suitable for publication now.